# Dataset: Traffic Images Captured from UAVs for Use in Training Machine Vision Algorithms for Traffic Management

Sergio Bemposta Rosende [1], Sergio Ghisler [2], Javier Fernández-Andrés [3] and Javier Sánchez-Soriano [4,*]

1 Department of Science, Computing and Technology, Universidad Europea de Madrid, 28670 Villaviciosa de Odón, Spain; sergio.bemposta@universidadeuropea.es
2 Stirling Square Capital Partners LLP, London SW3 4LY, UK; sghisler@gmail.com
3 Department of Industrial and Aerospace Engineering, Universidad Europea de Madrid, 28670 Villaviciosa de Odón, Spain; javier.fernandez@universidadeuropea.es
4 Higher Polytechnic School, Universidad Francisco de Vitoria, 28223 Pozuelo de Alarcón, Spain
* Correspondence: javier.sanchez@ufv.es

**Abstract:** A dataset of Spanish road traffic images taken from unmanned aerial vehicles (UAV) is presented with the purpose of being used to train artificial vision algorithms, among which those based on convolutional neural networks stand out. This article explains the process of creating the complete dataset, which involves the acquisition of the data and images, the labeling of the vehicles, anonymization, data validation by training a simple neural network model, and the description of the structure and contents of the dataset (which amounts to 15,070 images). The images were captured by drones (but would be similar to those that could be obtained by fixed cameras) in the field of intelligent vehicle management. The presented dataset is available and accessible to improve the performance of road traffic vision and management systems since there is a lack of resources in this specific domain.

**Dataset:** https://zenodo.org/record/5776219.

**Dataset License:** Creative Commons Attribution 4.0 International.

**Keywords:** dataset; UAV; intelligent vehicle; machine leaning; computer vision; convolutional neural network; model deployment; autonomous driving; roundabouts; deep learning; traffic management

## 1. Introduction

The use of UAVs began in the early 21st century for military purposes [1–3], but over time, they have started to be used in other sectors [4]. One of the most challenging problems in traffic control is complex maneuvers, such as at traffic roundabouts and intersections [5,6]. UAVs can be used to cooperate with autonomous vehicles [7] or traffic infrastructures [8] in these complex maneuvers, providing them with information that would not be available without such aerial vision [9].

Other works in the field of intelligent transport are the semantic segmentation of RGB-thermal images [10] as well as the identification of salient objects [11]. In these works, traditional RGB images are combined with thermal images as well as depth images to take advantage of all the available environmental information.

Neural networks have become the state-of-the-art technology for pattern detection [12], so it is easy to find a wide variety of networks for different purposes [13].

Other applications and uses of deep convolutional neural networks are proposed by the work in [14]. They work with stereoscopic images in three dimensions, but the need to use robust and complete datasets remains. Despite this, there is no model capable of detecting objects in images taken with UAVs or from fixed cameras installed at altitude, due, in part, to the difficulty of detecting small objects [15–17].

There are several datasets with aerial images captured from drones such as Vis-Drone [18], one of the most widely used. This dataset is composed of videos and images of all kinds of situations, totaling about 250,000 images of 11 different classes, including some vehicles. The images range in size from 1344 × 746 pixels to 2688 × 1512 pixels. All the images were captured in urban and highway environments in China. It does not contain roundabouts or split roundabouts, which are abundant in Europe and especially on Spanish roads. In addition, all of the signs and road signs are in Chinese.

For these reasons, the dataset described in this work has been created. This dataset will help in the training of neural networks and artificial intelligence systems in areas such as the identification of vehicles and other relevant actors in traffic management and intelligent vehicles. More specifically, this dataset can be used for several purposes, among which the most important are:

1. For the training of algorithms to monitor complex infrastructures such as roundabouts and junctions, etc. The information obtained can be used for V2V and V2I arbitration and communication systems currently under development in the automotive industry, which will improve safety in shared traffic.
2. For the training of algorithms that can identify different types of vehicles and can be implemented in autonomous UAVs used for traffic control and safety.
3. For the training of algorithms developed for the management of traffic violations monitored using UAVs, increasingly used by traffic agencies in many countries.
4. For the training of algorithms developed for emergency services to use UAVs for the rapid response to a traffic accident and to minimize the number of victims by the prompt assistance of emergency services.
5. For the training of algorithms developed for organizations in charge of designing and managing new road infrastructures to enable a better design of these infrastructures to improve safety and minimize traffic congestion.

## 2. Data Description

Our dataset is composed of 15,070 images in png format accompanied by as many files with txt extension with the description of the elements identified in each of the images. In total, there are 30,140 files comprising the images and descriptions. The images were taken in six different locations of urban and interurban roads, those with motorways being discarded. In these images, 155,328 vehicles have been labeled, including cars (137,602) and motorcycles (17,726). These data can be seen in Table 1 in more detail.

**Table 1.** Details of the obtained labels.

| Scenes | Frames | Targets | Cars | Motorbikes |
|---|---|---|---|---|
| Regional road | 4500 | 24,858 | 14,577 | 10,281 |
| Urban intersection | 2462 | 10,759 | 10,759 | 0 |
| Rural road | 1292 | 746 | 746 | 0 |
| Split roundabout | 2297 | 3107 | 3107 | 0 |
| Roundabout (far) | 1814 | 71,819 | 64,844 | 6975 |
| Roundabout (near) | 3997 | 44,039 | 43,569 | 470 |
| Total | 15,070 | 155,328 | 137,602 | 17,726 |

The dataset was created in You Only Look Once (YOLO) format [19] due to its widespread popularity, as well as the ease with which it can be adapted or converted to other formats due to its characteristics. In this format, the images and their annotations are named the same way (consecutive integer values, starting at 0) using the extension png

for the images and txt for the annotations associated with that image. In the txt files, the following notation is used to define five fields:

```
<object-class> <x> <y> <width> <height>
```

Specifically, each of the fields contains the following:
- Object class: Integer number varying between 0 and N-Classes-1. The two classes that have been incorporated in the model are as follows: 0. Cars, 1. Motorcycles.
- x, y: Decimal values relative to the center of the rectangle containing the labeled object. They vary in the range [0.0 to 1.0].
- Width, height: Decimal values relative to the width and height of the rectangle containing the labeled object. They vary in the range [0.0 to 1.0].

Figures 1 and 2 show the labeling of vehicles. In the case of the first one, it is a split roundabout with a complex situation. The second one was captured at a roundabout with a multitude of vehicles parked on the margins and some circulating inside.

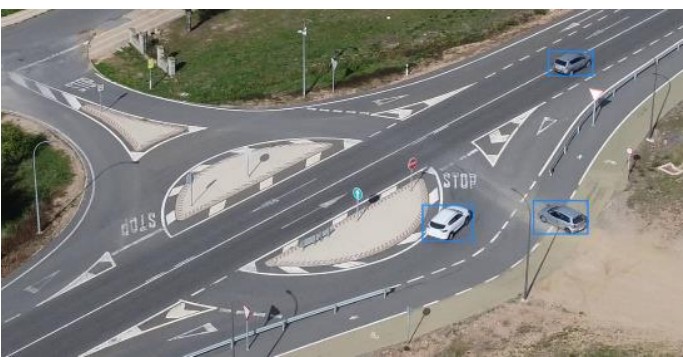

**Figure 1.** Three vehicles at a split roundabout.

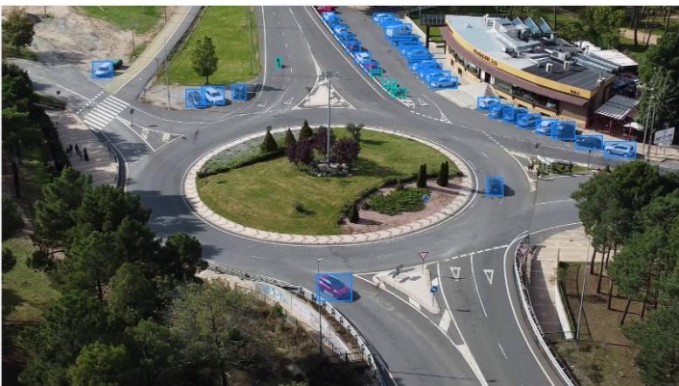

**Figure 2.** Roundabout with cars and motorcycles circulating around it, as well as a multitude of vehicles parked on the margins.

## 3. Methods

For the construction of the dataset, it was necessary to record videos to obtain our own image bank. According to the research in [1,20], which studied the requirements for recording a dataset of trajectories, we can extrapolate these requirements to the concrete objective of recognizing objects in images taken by drones.

The dataset must contain many images, as well as many labeled objects within these images. On the other hand, the images for model training must have been taken in different locations, with different visibility conditions to help the model avoid overfitting. Overfitting is one of the biggest obstacles in artificial intelligence; it occurs when the model learns in such a way that it can only be applied to the training dataset and, therefore, is not

generalizable to other data [21]. Finally, all types of objects must be recognized. When labeling images, objects that are related to the objects we want to predict should not be excluded; for example, we cannot exclude trucks from the dataset if we are labeling all cars; all objects can be included in a category such as "vehicles", or a category can be created for each of these objects.

With these requirements in mind, we proceed to describe each of the tasks performed for the construction of the dataset.

### 3.1. Obtaining the Dataset

The images of the dataset were taken by the authors using two different aircrafts, specifically the DJI Mavic Mini 2 UAVs and the Yuneec Typhoon H (Intel Real Sense). Details of the cameras used can be found in Table 2. The current regulations governing the civilian use of remotely piloted aircraft in Spain [22] were complied with. To capture the images, a series of video capture missions were planned, defining the locations, recording angles and orientations, and schedules, among other aspects. A wide range of data were obtained in terms of quantity and diversity of locations. Table 1 shows a breakdown of the photographs that were tagged in the dataset.

**Table 2.** Detail of the cameras and configuration used for image acquisition.

|  | Yuneec Typhoon H (CGO3) | DJI Mavic Mini 2 |
|---|---|---|
| Resolution | 1920 × 1080 px | 1920 × 1080 px |
| FOV angle | 98° | 83° |
| Focal (35 mm) | 14 mm | 24 mm |
| Aperture | f/2.8 | f/2.8 |
| Aspect ratio | 16:9 | 19:9 |
| Sensor | 1/2.3″ CMOS | 1/2.3″ CMOS |

### 3.1.1. Selection of Locations

Locations were defined considering their interest as well as the variety of scenarios, with complex junctions and intersections, such as roundabouts or split roundabouts, being of particular interest. In the same way, crossings and junctions with fast roads are also relevant due to their high accident rate. These scenarios have in common the possibility of recording vehicles from a wide range of angles, which allows the learning algorithms to be fed with greater variety. In addition, some straight road sections have also been incorporated where vehicles can be seen at a wide range of distances.

### 3.1.2. Angle and Orientation of the Recordings

The aircrafts used allow the angle of their cameras to be adjusted, which was set within the range 45–60° with respect to the horizontal axis of flight. This range allows one to capture the sides and the upper part of the targets, as opposed to a completely zenithal capture at 90° where only the upper part would be taken. The orientation chosen for the different captures was the one that provided the best framing of the scene compared to other criteria, such as maintaining the north orientation of the upper part of the images.

### 3.1.3. Height of the Recordings

The height of flight varied within the range 35–120 m, which corresponds to the minimum height recommended for safety and the maximum by regulation. This variation in heights allowed us to have a diversity of vehicle sizes and to provide variety to the dataset.

### 3.1.4. Relationship between Resolution and Recording Height

Camera resolution and flight altitude are related. The flight height was set according to the size of the objects to be detected, which is imposed by the average object size to be used for the subsequent object recognition algorithms.

The width of the scene in the images is 1920 pixels. If an element of known size is identified, the dimensions of the captured scene can be calculated. Figure 3 shows that a BMW X5 (4.86 m long and 1.78 m high) has 120 × 44 pixels in the image. If this image had been captured with the camera position at 90° to the horizontal, the flight height would be 77.76 m. However, as the scene is shot at an angle of 45° to the horizontal, the flight height is 55 m as can be seen in Figure 4.

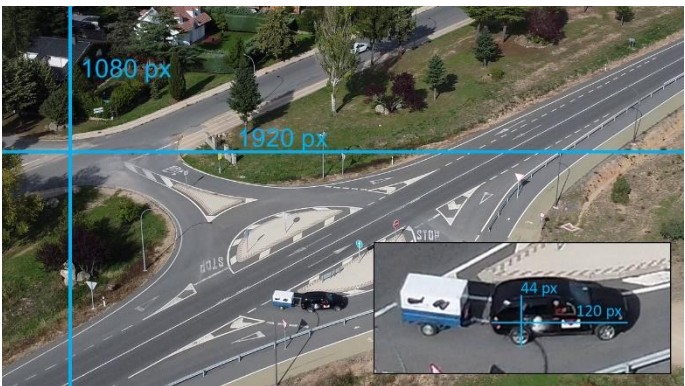

**Figure 3.** Frame resolution and vehicle size in pixels.

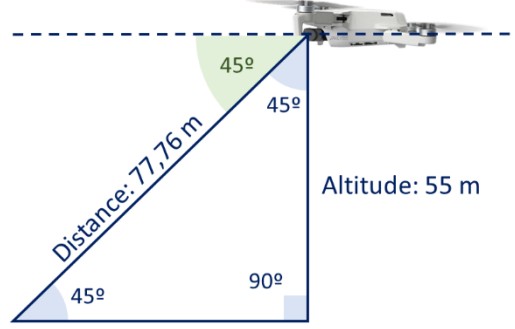

**Figure 4.** Camera angle and relation between distance and altitude.

### 3.1.5. Planning and Execution of the Missions

All recordings were carried out during the day and in good visibility conditions to comply with regulations. The recordings in interurban environments were carried out during the weekend, taking advantage of the fact that this is when most motorbikes are used for recreational purposes. The urban images were taken during the working week. The flights were carried out manually, with the pilot operating the aircraft controls rather than relying on automatic flight plans.

### 3.2. *Techniques for Anonymizing the Dataset*

To ensure compliance with European data protection regulations [23], the publication of images must comply with the requirement that they must not contain information that could identify individuals or lead to their identification through additional information. These images must be anonymized and, extending the term, pseudonymized. In our images, only the license plates of cars and people are sensitive information to be anonymized. In our dataset, there are no faces of people, so it was not necessary to apply tools in this sense. On the other hand, license plates are likely to be identifiable, so license plate deletion software filters have been developed.

For anonymization, we applied algorithms from the OpenCV library [24,25], but these algorithms focus on the process of "reading" the license plate once the image has been segmented and the region of interest (ROI) has been located. In our case, this is the most complex part since the portion of the license plate in the image is very small, so the segmentation can generate too much noise likely to be considered license plate. Figure 5 shows that since the license plates are small in proportion to the image, even discarding objects by size and proportion, there are many possibilities to be checked by the OCR.

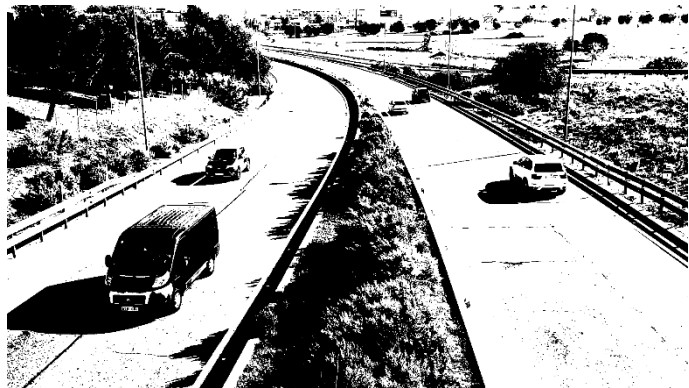

**Figure 5.** Binarized and thresholded image prepared for license plate segmentation.

After realizing that this was not the best option, we modified our approach to neural networks (deep learning networks), which are better adapted to locating number plates in an image (segmentation process). We used the public dataset "car plate detection" [26] available in Kaggle together with a selection of our own images from the dataset described in this paper to train a neural network YOLO V4 with a set of 500 images of European format license plates [27]. Once the image is segmented and the license plate is located, it is erased by a defocusing process. Figure 6 shows the result on the same image of the dataset, in which two license plates likely to be legible are located. In addition, there are three other vehicles in the image, but, due to their distance from the camera, the license plate cannot be detected, nor is it legible in the image.

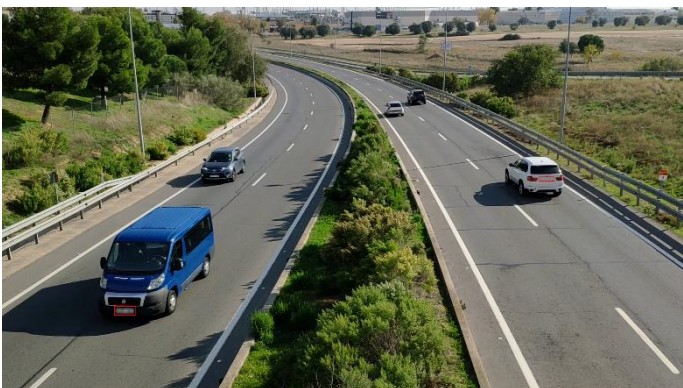

**Figure 6.** Localization and blurring of two license plates in a dataset image using neural networks.

### 3.3. Dataset Processing and Labeling

Once the images have been obtained, they must be processed and labeled to build models based on neural networks. The first step is to divide the videos obtained by the UAVs into images. For this, a Python script was developed, which, using the OpenCV computer vision library, allows us to choose a video and divide it into the number of images that compose it. The number of images that make up each video depends on the FPS (Frames Per Second) at which the video is recorded. The videos for the realization of

this work were captured at 30 FPS, so each second of the video is composed of 30 images. We eliminated the odd frames to reduce the size of the dataset as the changes between frames are not significant. There are different ways to label and detect objects in images [28,29]. For the development of the work, two methods were tested.

The first option was to detect objects with segmentation, but after the test, there was a problem in that since the objects were small and had a lot of background (everything that is not an object), the model had a 99% detection accuracy, simply by assigning the entire image as the background, as can be seen in Figure 7. We decided to use bounding boxes, rectangles that mark the boundaries of the object, since the metrics used in this type of annotation are not influenced by the background, but by the overlapping of the real rectangle with the inferred rectangle [29,30].

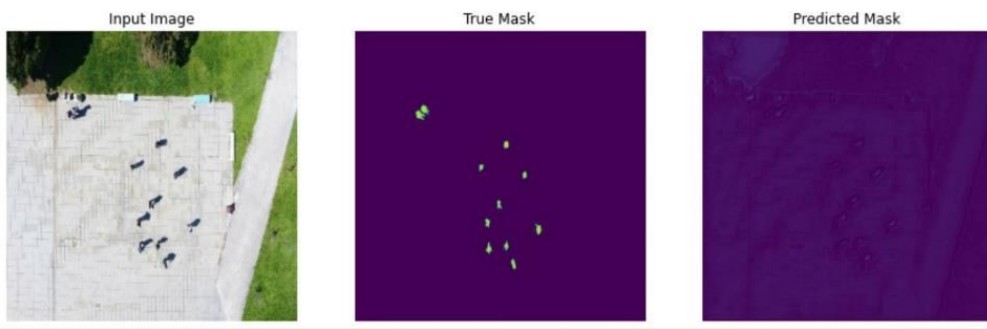

**Figure 7.** Model overestimation (99% accuracy) by detecting everything as background.

The CVAT tool was used to tag the images. This tool allows us to label images in a simple and agile way, owing to its "follow object" functionality, which allows us to label two images separated in time, and calculates, in an approximate way, the position of the object between these two images; to label accurately, an amplitude between images greater than 20 should not be exceeded, and it will be manually checked that the tracking has been performed correctly [31].

CVAT has a web application, but for greater security and to be able to modify default options such as the download size limit of the dataset once downloaded, we chose to deploy the application in containers owing to Docker (an open-source tool that allows virtualization in containers as if it were a virtual machine, but lightweight, thus enabling the automation of the deployment of these applications) [32].

Once the images are labeled and the dataset validated, they are exported to YOLO format, which is one of many export formats CVAT offers, and made available to the community to be used for model training.

### 3.4. Dataset Validation

The dataset was used to create a basic model to analyze its value. The selected model was yolov5m, a 365-layer PyTorch neuronal network. The mean average precision (mAP), with a 0.5 intersection over union (IoU) was established as the parameter to be optimized. Most of the images without objects (90%) were removed for this training to ensure the integrity of the results [33]. Tables 3 and 4 show the hardware used, the training parameters, and the result obtained, respectively. Figure 8 graphically illustrates the training results for the parameters mAp, loss (training set), and val_loss (test set).

**Table 3.** Hardware used for training.

| | |
|---|---|
| Processor | Intel Core i5-6500TE 2.4 GHz |
| Operating system | Ubuntu 20.04.3 LTS (Focal Fossa) |
| Motherboard | Intel RUBY-D718VG2AR |
| RAM | 64 GB |
| Graphics card | Nvidia RTX 2060 |
| Hard disk | 512 GB SSD |

**Table 4.** Training parameters and result.

| | |
|---|---|
| Intersection over Union: | 0.5 |
| Learning rate | 0.01 |
| Train/validation split | 90–10% |
| Steps | 3659 |
| Batch size | 4 |
| Total epochs | 55 |
| Total time training | 21.8 h |
| mAP | 0.97946 |
| Class car [mAP] | 0.994 |
| Class motorcycle [mAP] | 0.962 |
| Precision | 0.98456 |
| Recall | 0.95508 |

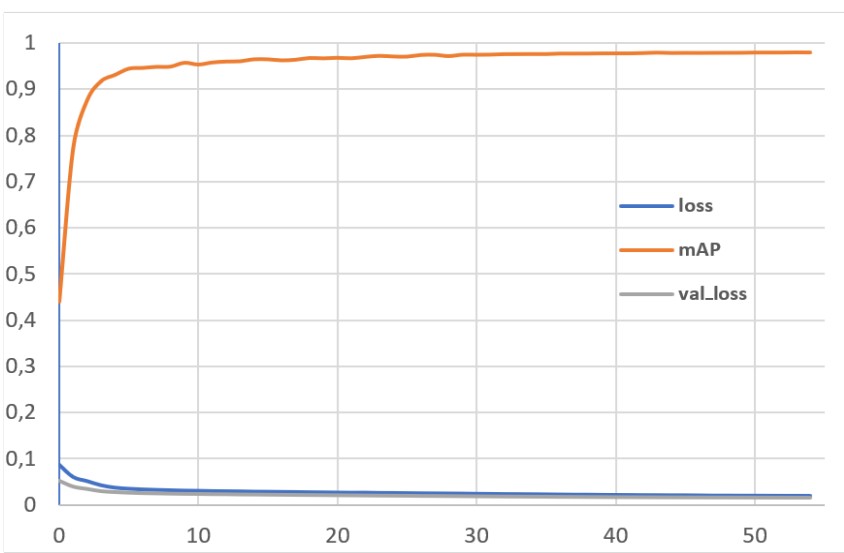

**Figure 8.** Results of training a basic model.

### 4. User Notes

The approach used to build the dataset shown in this work seeks to provide resources to train neural networks for artificial intelligence systems to be used in the identification of vehicles in traffic management.

**Author Contributions:** Conceptualization, J.S.-S., S.G. and S.B.R.; methodology, J.S.-S. and S.G.; software, S.G.; validation, J.S.-S., S.G. and S.B.R.; formal analysis, J.S.-S. and S.G.; investigation, J.S.-S. and S.G.; resources, J.S.-S., S.G. and S.B.R.; data curation, S.G.; writing—original draft preparation, J.S.-S. and S.G.; writing—review and editing, J.S.-S., S.G., S.B.R. and J.F.-A.; visualization, S.G. and S.B.R.; supervision, J.S.-S.; project administration, J.S.-S.; funding acquisition, J.F.-A. All authors have read and agreed to the published version of the manuscript.

**Funding:** This publication is part of the I+D+i projects with reference PID2019-104793RB-C32, PIDC2021-121517-C33, funded by MCIN/AEI/10.13039/501100011033/, S2018/EMT-4362/"SEGVAUTO4.0-CM" funded by Regional Government of Madrid and "ESF and ERDF A way of making Europe".

**Institutional Review Board Statement:** Not applicable.

**Informed Consent Statement:** Not applicable.

**Data Availability Statement:** The data presented in this study are openly available in https://zenodo.org/record/5776219 (accessed on 24 March 2022) with doi: https://doi.org/10.5281/zenodo.5776218.

**Conflicts of Interest:** The authors declare no conflict of interest.

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
