# Peer review of "Dataset: Traffic Images Captured from UAVs for Use in Training Machine Vision Algorithms for Traffic Management"

_data_

Round 1
Reviewer 1 Report
- The data collection process could be more specific. For example, how to record the locations, orientations and so on? What impact do these factors have on the data?
- The data set anonymization technique was well described.
- The results of using Bounding boxes should be showed to compare the first option.
Author Response
Dear reviewer,
We would like to thank you for taking the time to read and review the article. Please allow us to explain the changes made based on your comments:
Point 1: The data collection process could be more specific. For example, how to record the locations, orientations and so on? What impact do these factors have on the data?
Response 1: Thank you for allowing us to fill this gap.
Section "3.1 - Obtaining the data set" has been expanded to include information on the cameras used, the selection of locations, the recording angle and orientation of the drone, as well as the flight altitude. It has also been explained how the missions have been developed. In each of the cases, it has been indicated how these decisions may have influenced the results obtained.
Point 2: The data set anonymization technique was well described.
Response 2: The authors are grateful with this comment.
Point 3: The results of using Bounding boxes should be showed to compare the first option.
Response 3: Thank you very much for your comment.
Indeed, the results of the followed strategy were missing and the results of the discarded strategy based on segmentation were shown.
Section "3.4. Dataset validation" has been added, where the performance is calculated. For this purpose, our dataset is trained and tested using the YOLO 5 algorithm in order to validate and obtain a measure of its performance. A new reference have been introduced in this section.

Reviewer 2 Report
This article explains the process of creating the complete dataset, from the acquisition of the data and images, the labelling of the vehicles and the description of the structure and contents of the dataset itself (which amounts to 15.070 images). However, in my opinion, several concerns listed as follows should be clarified or revised.
- I suggest applying some of the latest objective algorithms to your proposed data set and comparing the performance of the various algorithms.
- 2. It is suggested that the relevant work survey scope of intelligentvehicles , not only limited to unmanned aerial vehicles (UAV), (e.g. MFFENet, GMNet, IRFR-Net), so as to bring more inspiration to this study and readers.
- Related work is a bit inadequate. Other related methods should be mentioned and analyzed in the related section of this paper. For example,Salient object detection in stereoscopic 3D images using a deep convolutional residual autoencoder.
Author Response
Dear reviewer,
We would like to thank you for taking the time to read and review the article. Please allow us to explain the changes made based on your comments:
Point 1: I suggest applying some of the latest objective algorithms to your proposed data set and comparing the performance of the various algorithms.
Response 1: Thank you very much for your comment.
Section "3.4. Dataset validation" has been added, where the performance is calculated. For this purpose, our data set is trained and tested using the YOLO 5 algorithm in order to validate and obtain a measure of its performance. A new reference have been introduced in this section.
In the team we intend to use the data set to train different algorithms as proposed by the reviewer, in order to compare their performance. We also intend to validate their usefulness by running them on different reduced-board equipment onboard UAVs (or fixed cameras). But these experiments are intended to be developed in detail in another work in the authors' line of research, once we have this consolidated dataset.
Point 2: It is suggested that the relevant work survey scope of intelligentvehicles , not only limited to unmanned aerial vehicles (UAV), (e.g. MFFENet, GMNet, IRFR-Net), so as to bring more inspiration to this study and readers.
Response 2: Thank you very much for the idea.
References to GMNet, IRFR-Net have been introduced in the introduction, for a better context of the work in the area of "Intelligent vehicles".
This term "Intelligent vehicles", has been included in the keywords and is also introduced in the Abstract, where it is explained that the images have been captured with drones but that could be used for other purposes associated with fixed traffic or surveillance cameras.
The “Summary” chapter is renamed “Introduction” to better contextualise the changes.
Point 3: Related work is a bit inadequate. Other related methods should be mentioned and analyzed in the related section of this paper. For example,Salient object detection in stereoscopic 3D images using a deep convolutional residual autoencoder.
Response 3: Thank you very much for your comment.
A reference to work "Salient object detection in stereoscopic 3D images using a deep convolutional residual autoencoder" is introduced in order to extend the related work. In this sense, it should be noted that this section has been generally improved with the addition of the two references mentioned in the previous answer (GMNet, IRFR-Net).

Reviewer 3 Report
The value of theory and application is limited。
Author Response
Dear reviewer,
We would like to thank you for taking the time to read and review the article. Please allow us to explain the changes made based on the comments of all reviewers.
Point 1: The value of theory and application is limited.
Response 1: The following changes and improvements have been made to the manuscript:
- Errors and misprints are corrected.
- “Summary” chapter is renamed “Introduction” to better contextualise the changes.
- New references (3) have been introduced in the introduction, for a better context of the work in the area of "Intelligent vehicles".
- Term "Intelligent vehicles" has been included in the keywords and is also introduced in the Abstract, where it is explained that the images have been captured with drones but that could be used for other purposes associated with fixed traffic or surveillance cameras.
- Section "1 - Obtaining the data set" has been expanded to include information on the cameras used, the selection of locations, the recording angle and orientation of the drone, as well as the flight altitude. It has also been explained how the missions have been developed. It has been indicated how these decisions may have influenced the results obtained.
- Section "3.4. Dataset validation" has been added, where the performance is calculated. For this purpose, our dataset is trained and tested using the YOLO 5 algorithm in order to validate and obtain a measure of its performance. A new reference have been introduced in this section.

Reviewer 4 Report
Authors, in their paper, present a dataset of road traffic images taken from unmanned aerial vehicles (UAV). The purpose of this dataset is to train artificial vision algorithms, among which those based on convolutional neural networks. This article explains the process of creating the complete dataset, from the acquisition of the data and images, the labeling of the vehicles, and the description of the structure and contents of the dataset itself. The authors present an interesting procedure to create a dataset by means of UAVs and the careful preparation of a publicly available dataset. However, the paper can be improved. Some comments and suggestions are the following:
_ Table 1, Table caption has to be located above the table. Please, correct.
_ Authors should explain in detail the procedure that they follow to create the different missions, locations' definition, recording angles and orientations, schedules, etc. Please describe this procedure.
_ How do authors determine UAV location? If a UAV hovers in a “static” location, how do they handle the error of small movements? Which is the resolution of cameras used? Which is according to their opinion the optimum camera's resolution depending on the desired altitude? Please explain.
_ Please, describe the procedure to export images to YOLO format. Add this description to your text.
_The preparation of a dataset that can be used for training is generally a difficult and time-consuming procedure. On the other side, reliable and accurate results in machine learning require as more as possible properly selected data. How do authors ensure that according to their approach to prepare the dataset, create an optimum dataset? Do they implement some specific criteria to select for example acquisition location, angle, etc? Please explain.
Author Response
Dear reviewer,
We would like to thank you for taking the time to read and review the article. Please allow us to explain the changes made based on your comments:
Point 1: Table 1, Table caption has to be located above the table. Please, correct.
Response 1: Thank you. The title of Table 1 has been placed above it.
Point 2: Authors should explain in detail the procedure that they follow to create the different missions, locations' definition, recording angles and orientations, schedules, etc. Please describe this procedure.
Response 2: Thank you for allowing us to fill this gap.
Section "3.1 - Obtaining the data set" has been expanded to include information on the cameras used, the selection of locations, the recording angle and orientation of the drone, as well as the flight altitude. It has also been explained how the missions have been developed. In each of the cases, it has been indicated how these decisions may have influenced the results obtained.
Point 3: How do authors determine UAV location? If a UAV hovers in a “static” location, how do they handle the error of small movements? Which is the resolution of cameras used? Which is according to their opinion the optimum camera's resolution depending on the desired altitude? Please explain.
Response 3: Thank you very much for your comment. We proceed to answer the questions:
How do authors determine UAV location? The location of the UAV has been determined to achieve a suitable framing of the desired location. These criteria are mentioned in response 2 and have been incorporated into the manuscript.
If a UAV hovers in a “static” location, how do they handle the error of small movements? Small changes due to GPS accuracy or inclement weather (mainly wind) affect the location in cases where "static" catches are intended. No additional measures have been taken to handle these events, which are considered to be acceptable and not very significant for the results.
Which is the resolution of cameras used? Full HD resolution (1920 x 1080 pixels) has been selected for both aircraft. A new table (Table 2) is included with the information on the cameras used.
Which is according to their opinion the optimum camera's resolution depending on the desired altitude? The camera resolution is defined in each drone by its camera, which cannot be changed. In the case of the DJI Mavic Mini 2 UAV, the camera can capture images from 1920x1080 pixels up to 4000x3000 pixels. On the Yuneec Typhoon H UAV the camera resolution is only 1920x1080 pixels. So that all images are the same, the same resolution has been selected for both devices (1920x1080). The height of flight is conditioned by the minimum resolution to capture the objects in the images. The relationship between height and width of the scene in the DJI UAV is: flight height = scene width/1.77. In the Yuneec UAV it is: flight height = scene width/2.22.These ratios have been obtained with the parameters of the optics of both cameras described in the article. This allows the Yuneec UAV to fly lower than the DJI UAV for the same image object resolution. This information has been used for flight height calculations in a concrete case as shown in the article (figures 3 and 4).
Point 4: Please, describe the procedure to export images to YOLO format. Add this description to your text.
Response 4: Thank you very much for your comment.
Since we used CVAT for the labeling, this proccess was really simple, as YOLO is one of many formats CVAT allows users to export your dataset. This clarification is added to the manuscript.
Point 5: The preparation of a dataset that can be used for training is generally a difficult and time-consuming procedure. On the other side, reliable and accurate results in machine learning require as more as possible properly selected data. How do authors ensure that according to their approach to prepare the dataset, create an optimum dataset? Do they implement some specific criteria to select for example acquisition location, angle, etc? Please explain.
Response 5: Thank you very much for your comment.
Section "3.4. Dataset validation" has been added, where the performance is calculated. For this purpose, our dataset is trained and tested using the YOLO 5 algorithm in order to validate and obtain a measure of its performance. A new reference have been introduced in this section.
As can be seen, the results are good and justify the design of the dataset preparation.
Criteria have indeed been used for the design of the dataset. Details on these criteria have been introduced in section 3.1, as explained in response 2.

Round 2
Reviewer 2 Report
The author carefully revised the paper and addressed my concerns in detail. Thus, I recommend accepting this paper.
Author Response
Dear reviewer,
We would like to thank you for taking the time to read and review the article. Please allow us to explain the changes made based on your comments:
Point 1: The author carefully revised the paper and addressed my concerns in detail. Thus, I recommend accepting this paper.
Response 1: The authors are grateful with this comment. The improvements you proposed in the last round allowed us to improve the manuscript considerably.

Reviewer 4 Report
Authors, in their paper, present a dataset of road traffic images taken from unmanned aerial vehicles (UAV). The purpose of this dataset is to train artificial vision algorithms, among which those based on convolutional neural networks. This article explains the process of creating the complete dataset, from the acquisition of the data and images, the labeling of the vehicles, and the description of the structure and contents of the dataset itself. The authors present an interesting procedure to create a dataset by means of UAVs and the careful preparation of a publicly available dataset. The paper has been improved. Some suggestions are the following:
_ Table 1, Table caption has to be located above the table. Please, correct again this caption.
_ A few typos exist in the text (lines 148, 191, 203, OpenCV is one word, 251). Please correct.
_ Figure 7. Which is the meaning of the words loss and val? Does not exist a reference in the text. Please correct.
Author Response
Dear reviewer,
We would like to thank you for taking the time to read and review the article. Please allow us to explain the changes made based on your comments:
Point 1: Table 1, Table caption has to be located above the table. Please, correct again this caption.
Response 1: Thank you. The title of Table 1 was placed above the table in the last revision.
For some reason, it may not display properly due to change control or being converted to PDF by the application, but it was indeed changed as you indicated in the last round. This can be checked in the attached word document.
Point 2: A few typos exist in the text (lines 148, 191, 203, OpenCV is one word, 251). Please correct.
Response 2: Thank you very much for your comment. All typos have been corrected.
Point 3: Which is the meaning of the words loss and val? Does not exist a reference in the text. Please correct.
Response 3: Thank you very much for your comment.
In Figure 7, "val/loss" has been changed to the correct name of the value "val_loss". A reference has also been included in the text. These values (mAp, loss and val_loss) are commonly used to verify training results in Keras.
